# Direct Interaction between *N*-Acetylcysteine and Cytotoxic Electrophile—An Overlooked In Vitro Mechanism of Protection

**DOI:** 10.3390/antiox11081485

**Published:** 2022-07-29

**Authors:** Petr Mlejnek

**Affiliations:** Department of Anatomy, Faculty of Medicine and Dentistry, Palacky University Olomouc, Hnevotinska 3, 77515 Olomouc, Czech Republic; petr.mlejnek@upol.cz

**Keywords:** *N*-acetylcysteine, nucleophile, electrophile, *N*-acetylcysteine-electrophile adduct, mechanism of protection

## Abstract

In laboratory experiments, many electrophilic cytotoxic agents induce cell death accompanied by reactive oxygen species (ROS) production and/or by glutathione (GSH) depletion. Not surprisingly, millimolar concentrations of *N*-acetylcysteine (NAC), which is used as a universal ROS scavenger and precursor of GSH biosynthesis, inhibit ROS production, restore GSH levels, and prevent cell death. The protective effect of NAC is generally used as corroborative evidence that cell death induced by a studied cytotoxic agent is mediated by an oxidative stress-related mechanism. However, any simple interpretation of the results of the protective effects of NAC may be misleading because it is unable to interact with superoxide (O_2_•^−^), the most important biologically relevant ROS, and is a very weak scavenger of H_2_O_2_. In addition, NAC is used in concentrations that are unnecessarily high to stimulate GSH synthesis. Unfortunately, the possibility that NAC as a nucleophile can directly interact with cytotoxic electrophiles to form non-cytotoxic NAC–electrophile adduct is rarely considered, although it is a well-known protective mechanism that is much more common than expected. Overall, apropos the possible mechanism of the cytoprotective effect of NAC in vitro, it is appropriate to investigate whether there is a direct interaction between NAC and the cytotoxic electrophile to form a non-cytotoxic NAC–electrophilic adduct(s).

## 1. Introduction

*N*-acetylcysteine (NAC), the acetylated derivative of the amino acid L-cysteine, serves as a safe agent in many clinical settings. Originally it was used as a mucolytic agent to alleviate symptoms in patients with cystic fibrosis or in those with a variety of respiratory illnesses. NAC is clinically effective in the treatment of paracetamol overdose or poisoning as well as having clinical value in acute heavy metal poisoning. NAC was also experimentally used for the treatment of disorders associated with glutathione (GSH) depletion and/or reactive oxygen species (ROS) production, including diabetes, cancer, neuropsychiatric disorders, human immunodeficiency virus, and many others [1,2,3,4,5]. Newer review articles on the use of NAC in clinical practice can be found elsewhere [6,7,8].

In laboratory experiments, NAC is the most commonly used protective agent against cytotoxic electrophiles, oxidants, and metals. Although it is a reactive compound that enters into many chemical reactions, its protective properties are most often attributed to its ability to scavenge ROSs directly or indirectly through stimulation of intracellular GSH synthesis. However, protective properties can be also ascribed to its nucleophilicity, which blocks the cytotoxicity of reactive electrophiles [9,10]. In addition to the three common protective mechanisms listed above, NAC can exert cytoprotective effects either on conversion to sulphan sulphur species in mitochondria or by modulation of signalling pathways through redox-sensitive transcription factors [11,12]. Moreover, NAC can efficiently eliminate the cytotoxicity of many metals by chelation [4,9,13]. An overview of the possible mechanisms by which NAC eliminates the effect of cytotoxic agents is given in Figure 1.

Although the direct non-enzymatic reaction of NAC with cytotoxic electrophiles to form corresponding non-cytotoxic adducts is a broadly applicable mechanism of protection, it is often overlooked. In contrast, the protective effects of NAC are often unjustifiably attributed to its ability to directly or indirectly scavenge ROS. In this review article, I will try to provide readers with arguments as to why I think this is the case, and I will also suggest ways to avoid erroneous conclusions about the mechanism of NAC’s protective effect.

## 2. Reactive Oxygen Species

### 2.1. Definition of Reactive Oxygen Species

Reactive oxygen species are highly reactive molecules comprising radicals, ions, or molecules with a single unpaired electron in their outermost shell. ROSs are divided into two groups: radical and non-radical. The first includes superoxide (O_2_•^−^), hydroxyl radical (•OH), alkoxyl radicals (RO•), peroxyl radicals (ROO•), sulphonyl radicals (ROS•), and thiyl peroxyl radicals (RSOO•). The second includes hydrogen peroxide (H_2_O_2_), organic hydroperoxides (ROOH), hypochloride (HOCl), ozone/trioxygen (O_3_), and singlet oxygen (^1^O_2_). Detailed chemical properties of these ROSs are given elsewhere [4,14]. 

It should be noted that other radicals, ions, or molecules with a single unpaired electron also exist. Reactive nitrogen species (RNS) include nitric oxide (NO•), peroxynitrite (ONO^−^), nitrosoperoxycarbonate anion (O=NOOCO_2_^−^), nitrocarbonate anion (O_2_NOCO_2_^−^), dinitrogen dioxide (N_2_O_2_), nitronium (NO^2+^), organic radicals (R•), thiyl radicals (RS•), and highly reactive lipid- or carbohydrate-derived carbonyl compounds, to name the most important ones [4,14]. In this review, however, I will focus only on the following ROSs: O_2_•^−^, H_2_O_2_, and •OH, whose increased production may be induced by the action of some cytotoxic electrophiles on cells.

### 2.2. Sources of Reactive Oxygen Species

ROSs are generated as products of physiological processes such as mitochondrial respiration, oxidative reactions in peroxisomes, catalytic cycling of cytoplasmatic oxidoreductases (e.g., xanthine oxidase), hormone or growth factor-mediated ROS production (e.g., serotonin-smooth muscle cells), host-defence function of phagocytic cells (e.g., NADPH oxidase), and others. Increased production of ROSs is also associated with pathophysiological conditions (e.g., ischemia/reperfusion, cancer, cardiovascular disease, diabetes, etc.). Alternatively, some exogenous noxa (e.g., cytotoxic chemicals, transient metals) may induce excessive ROS production [15,16]. Detailed schematics of the cellular pathways of ROSs’ generation and their interactions are presented in a comprehensive review by Valko et al. [16].

### 2.3. Biological Effects of Reactive Oxygen Species

ROSs are involved in various intracellular physiological and pathophysiological processes. Traditionally, they are associated with negative effects. Nevertheless, in recent decades, low levels of ROSs have been found to serve both as signalling molecules and play an important role in regulating essential processes such as cell proliferation and differentiation. For example, some hormones, growth factors, and cytokines can generate low to mild ROS production in ligand-receptor-associated signalling pathways that regulate proliferation in nonphagocytic cells [17,18,19]. Neutrophils and macrophages can produce high levels of ROSs by reactions involving NADPH oxidases or myeloperoxidase on stimulation [20]. Last but not least, ROSs are involved in the regulation of cell death pathways for apoptosis, autophagy, necroptosis, pyrotosis, and ferroptosis [21,22]. 

Higher levels of ROSs are associated with damage to cell structures (e.g., membranes), organelles (e.g., mitochondria, lysosomes), and macromolecules (e.g., RNA, DNA, and proteins). For example, •OH damages pyrimidine and purine bases as well as the deoxyribose backbone, a crucial component of the DNA molecule. Such DNA damage, if irreparable, leads to mutations. Nuclear DNA mutations are involved in carcinogenesis, and mitochondrial DNA mutations are implicated in degenerative diseases [23,24]. In proteins, the side chains of the amino acid residues cysteine, methionine, arginine, histidine, tyrosine, proline, and glutamate are most vulnerable to damage by various ROSs. Damage of proteins by ROS is associated with a number of age-related diseases [25]. The involvement of increased ROS production in other physiological and pathophysiological processes is discussed in detail elsewhere [15,16,23]. 

### 2.4. Reactive Oxygen Species and Cellular Antioxidant Defence Mechanisms

Cells have developed defensive mechanisms for protecting themselves from the adverse effects of ROSs. These include different classes of antioxidants such as low molecular weight reducing molecules, thioredoxin, and enzymes which reduce ROS. 

Low molecular weight antioxidants include glutathione (GSH), lipoic acid, melatonin, and a number of molecules that are part of the diet such as vitamins C and E, carotenoids, flavonoids, and some others [26]. In addition, NADH and NADPH should be mentioned, although many researchers consider them indirect antioxidants acting as hydride donors in many processes catalysed by oxidoreductases. Reduction of oxidised glutathione (GSSG) to GSH can be given as an example. Importantly, Kirsch and De Groot proposed NADH and NADPH as directly operating antioxidants in the mitochondrial compartment [27]. 

Thioredoxin (TRX) is a small redox-active protein (12 kDa), reducing disulphide in various proteins including transcription factors, and is involved in the regulation of cell survival and proliferation [28]. Oxidised TRX is reduced by NADPH in a reaction catalysed by thioredoxin reductase (EC 1.8.1.9). 

The most important enzymes that reduce ROS (antioxidant enzymes) include superoxide dismutases (cytosolic SOD1, mitochondrial SOD2, and extracellular SOD3; EC 1.15.1.1), which catalyse the dismutation of O_2_•^−^ to H_2_O_2_ and O_2_, catalase (CAT; EC 1.11.1.6), which catalyses the conversion of H_2_O_2_ to H_2_O and O_2_, and glutathione peroxidases (GPx; EC 1.11.1.19), which in conjunction with glutathione (GSH) catalyse the reduction of H_2_O_2_ and/or organic peroxides to water or alcohols [29,30].

### 2.5. Reactive Oxygen Species and Antioxidants In Vitro

A number of antioxidants are used to scavenge ROSs and to prevent cell damage in in vitro experiments. Scientists prefer to use those that have the most universal properties with the potential to protect cells from the widest possible range of ROSs. 

GSH is considered as such a universal ROS scavenger and antioxidant. However, among biologically relevant ROSs, O_2_•^−^, H_2_O_2_, and •OH, GSH directly scavenges only •OH with high efficiency [31]. GSH is not a directly effective O_2_•^−^ scavenger and its non-enzymatic reduction of H_2_O_2_ takes place with low efficiency [32]. On the other hand, it is involved in the very efficient decomposition of peroxides (both H_2_O_2_ and ROOH), which occurs exclusively by the enzymatic action of GPx [32,33]. It should be pointed out that GSH is optimally active at an acidic pH as it becomes autooxidised to GSSG at a neutral and alkaline pH. A more detailed description of GSH’s cytoprotective effects is provided in chapter 3.2. 

Thiourea is non-cytotoxic and broad-spectrum ROS scavenger that interacts effectively with O_2_•^−^, H_2_O_2_, and •OH [34]. Despite this advantage, it is rarely used as a direct ROS scavenger to protect cells from death in laboratory experiments.

Ebselen is preferably used in laboratory experiments as a protective agent wherever elimination of the adverse effects of H_2_O_2_ is desired [35]. However, ebselen shows protective effects only at low concentrations. Conversely, at high concentrations, it can induce increased ROS production and cell death [36]. 

Ascorbic acid (vitamin C) serves as an excellent antioxidant interacting with O_2_•^−^ and H_2_O_2_ in vitro. Importantly, in the presence of metals such as iron it becomes a powerful source of ROS [37]. 

NAC and its derivative *N*-acetylcysteine amide (NACA) with improved membrane permeability [38] are frequently used as universal cytoprotective antioxidants which interact with a broad spectrum of ROSs and as precursors of GSH synthesis. However, as discussed in this review, NAC cannot be considered a universal ROS scavenger and its ability to stimulate GSH synthesis should be taken with caution. Other protective mechanisms need to be considered [4,9,13]. 

Chelating agents such as deferoxamine are preferably used as antioxidants in some experimental systems because, e.g., deferoxamine inhibits the Fenton reaction [37,39]. 

Dithiotreitol (DTT) and meracptoethanol (MeSH) are usually used in extraction and/or assay buffers to prevent oxidation of thiol protein groups. Their application as cytoprotective antioxidants is limited due to their cytotoxicity, particularly at high concentrations [40,41]. 

Trolox, a water-soluble analogue of vitamin E, can efficiently “repair” free radicals derived from amino acids including, tryptophan, tyrosine, histidine, and methionine, i.e., “repair” the free radical-oxidised proteins [42,43]. 

Frequently used antioxidants in laboratory experiments are listed in Table 1. Some food constituent antioxidants such as flavonoids, carotenoids, etc., are not listed. Although these have wide-ranging antioxidant activity, their application in in vitro experiments is limited [44]. 

## 3. Protective Mechanisms of NAC In Vitro

### 3.1. NAC as a Direct ROS Scavenger In Vitro

In laboratory experiments, NAC can counteract many electrophilic agents whose cytotoxicity is accompanied by ROS production and/or by GSH depletion. NAC’s protective effects are often ascribed to its ability to directly scavenge ROS in laboratory experiments (Figure 2a). Importantly, even though NAC is considered a universal ROS scavenger, data derived from the literature do not support this generally accepted belief [4,5,45,46,47]. Thus, NAC serves as an excellent scavenger of hypochlorous acid (HOCl) and hydroxyl radicals (•OH), however, it interacts only slowly with hydrogen peroxide (H_2_O_2_) and its interaction with superoxide (O_2_•^−^), the most important biologically relevant ROS, is not detectable [4,44]. Apropos thiol reactivity with O_2_•^−^, there is some controversy. Nevertheless, most authors would probably agree that NAC is a weakly reactive thiol either with O_2_•^−^ or H_2_O_2_ [28,42,43]. Therefore, in experimental systems with elevated O_2_•^−^ or H_2_O_2_ production, the protective properties of NAC can be hardly attributed to its ability to directly scavenge ROSs.

For the sake of completeness, it should be added that NAC can extensively stimulate ROS production under certain experimental conditions. For example, Zheng et al. demonstrated that NAC interacts with Cu^2+^ to generate H_2_O_2_, which induces cell death in cancer cells [48]. Similarly, we observed that NAC at low millimolar concentrations significantly enhanced the cytotoxic effect of Cd^2+^ through extensive ROS production [49]. In addition, we recently reported that NAC alone can induce a massive production of ROSs, resulting in loss of cell viability in human leukaemia cells [50].

### 3.2. NAC as an Indirect ROS Scavenger In Vitro

The indirect protective effects of NAC against the cytotoxic effects of electrophiles consist in stimulation of GSH biosynthesis (Figure 2b). This indirect protection mechanism is very easy to understand in light of the following facts. 

First, NAC serves as a precursor of glutathione (GSH) biosynthesis that takes place in the cytosol of cells in two steps. The first step, catalysed by glutamate cysteine ligase (GCL; EC 6.3.2.2), involves the formation of a γ-peptide bond between glutamic acid and cysteine to form γ-glutamylcysteine. In the second, step catalysed by GSH synthetase (GS, EC 6.3.2.3), the peptide bond between the carboxyl of cysteine and the amine of glycine is created to form GSH. The first step is rate limiting and regulated by the availability of L-cysteine [51,52]. NAC on deacetylation may unblock this restriction step [5]. 

Second, intracellular GSH levels fundamentally determine the elimination efficiency of superoxide and H_2_O_2_ in cells. Indeed, GSH serves as a co-substrate for the elimination of superoxide in the superoxide dismutase (SOD)-GPx coupled enzymatic reaction and co-substrate for elimination of H_2_O_2_ and organic peroxides in an enzymatic reaction catalysed by GPx [53]. 

For this reason, if addition of NAC in experiments is found to prevent GSH depletion, ROS production, and cell death, the logical conclusion is that NAC acts as an indirect ROS scavenger (Figure 2b). However, careful evaluation of all aspects of the experiment is necessary when interpreting the results. For example, the NAC concentration needed to achieve protection must be analysed. If very high concentrations (4–10 mM) are required, the mechanism of the protection may be different. Indeed, laboratory results show that even submillimolar concentrations of NAC significantly stimulate GSH synthesis [54,55]. 

### 3.3. NAC as a Nucleophile Blocks the Cytotoxicity of Electrophiles In Vitro 

The nucleophilicity of NAC is well known to counteract the cytotoxicity of reactive electrophiles and is therefore proposed as a third possible protective mechanism in vitro [9,10]. Although the direct reaction between the nucleophilic NAC and the electrophilic cytotoxic compound is fundamentally different, it is accompanied by the same external manifestations: replenishment of GSH, inhibition of ROS formation, and prevention of cell death. The outstanding difference is the mechanism of the GSH depletion. GSH as a nucleophile reacts directly with electrophiles to form non-cytotoxic adducts (Figure 3). In addition, GSH can be conjugated with a cytotoxic electrophile by a glutathione S-transferase (GSTs*;* EC *2*.5.1.18) catalysed reaction [56]. Both non-enzymatic and enzymatic interactions between GSH and a cytotoxic electrophile can cause GSH depletion (Figure 3). Depletion of GSH then leads to increased ROS production (Figure 3). The real protective effect of NAC is that the extracellular formation of the NAC–electrophile adduct dramatically reduces the concentration of the “free” electrophile, preventing intracellular interaction with GSH which obviates its depletion and all the related negative consequences (Figure 4). Alternatively, the cytotoxic electrophile may itself contribute to increased ROS production (Figure 3). Because excessive NAC (millimolar concentrations) is usually used relative to the amount of cytotoxic electrophile (micromolar concentrations), the conversion of the electrophile to the adduct is quantitative. 

The latter mechanism by which NAC can block the cytotoxic effects of a number of electrophilic substances undoubtedly plays a very important role as organic chemistry lessons show [57,58,59]. However, the published results suggest the opposite. To our knowledge, only a few reports demonstrate that NAC’s protective effects are related to its nucleophilicity. These studies are discussed in more detail in the following chapters.

#### 3.3.1. NAC and Patulin Cytotoxicity Prevention

Patulin (PAT; 4-hydroxy-4*H*-furo[3,2-*c*]pyran-2(6*H*)-one) is a mycotoxin that can be found in food, specifically in mouldy fruits and vegetables [60]. The first studies showed its acute organ toxicity in experimental animals [61]. In vitro experiments on mammalian cells have also shown that PAT is a mutagenic and carcinogenic agent. However, there is no general agreement on this topic [60]. It has also been shown that the cytotoxic effects of PAT are accompanied by increased ROS production and GSH depletion and cell death was attributed to an oxidative stress-related mechanism [62]. Contrary to this original hypothesis, PAT was thought to exert its cytotoxicity primarily by forming covalent bonds with the sulphydryl groups of proteins and GSH [63]. The chemical structures of the adducts that result from the non-enzymatic reaction between PAT and GSH or other low molecular weight thiols were soon identified and characterised [64]. Importantly, it was observed that a major adduct pattern formed between PAT and GSH was the same structural type as that obtained with NAC [64]. Recent laboratory research emphasises that PAT induces increased ROS production, GSH depletion, and ultimately cell death with apoptotic features [65]. Further, since the addition of NAC to the growth medium prevents all the adverse effects of PAT, including cell death, many authors explain the cytoprotective effects of NAC against the cytotoxic effects of PAT by its ability to scavenge the ROS [66,67] and/or by repletion of GSH [68]. Unfortunately, the direct and quantitative interaction between the electrophilic PAT and nucleophilic NAC and GSH has been almost forgotten.

#### 3.3.2. NAC and Cytotoxicity of 18beta-Glycyrrhetinic Acid Derivatives

The cytotoxic effects of 18beta-glycyrrhetinic acid (GA) derivatives, methyl 2-cyano-3,11-dioxooleana-1,12-dien-30-oate (CDODO-Me-11) and methyl 2-cyano-3,12-dioxooleana-1,12-dien-30-oate (CDODO-Me-12) were studied in human leukaemia cells [69]. Cell death induced by CDODO-Me-11 and CDODO-Me-12 was associated with complex changes in proteins regulating apoptosis, including downregulation of c-FLIP, XIAP, and Mcl-1 that precede cell death with morphological and biochemical features of apoptosis. In addition, apoptotic cell death induced by CDODO-Me-11 and CDODO-Me-12 was accompanied by depletion of GSH [69]. Even though the cytotoxic effects of CDODO-Me-11 and CDODO-Me-12 were abolished by NAC and GSH, the authors were not satisfied with a simple explanation that cell death is caused by GSH depletion. Instead, they performed further experiments to show that both NAC and GSH directly interact with CDODO-Me-11 and CDODO-Me-12 to form non-cytotoxic adducts [69]. 

#### 3.3.3. NAC and Cytotoxicity of Electrophiles Studied in Our Lab

Recent results from our laboratory are examples where older interpretations explaining the protective effects of NAC against the cytotoxic effect of geldanamycin (GDN), carbonyl cyanide 4-(trifluoromethoxy)phenylhydrazone (FCCP), and 3,5-bis[(2-fluorophenyl)methylene]-4-piperidinone (EF-24) have been completely revised based on new findings. We presented new evidence that NAC nucleophilicity rather than its direct or indirect potential for ROS scavenging and/or repletion of GSH prevented GDN, FCCP, and EF-24-induced cell death [55,70,71]. 

##### NAC and Cytotoxicity of GDN

GDN is a benzoquinone ansamycin antibiotic with strong antiproliferative activity against cancer cells [72,73,74]. The anticancer activity of GDN exerted at low micromolar concentrations (0.5–4 μM) is due to the inhibition of heat shock protein 90 (HSP90), a molecular chaperone that folds and stabilises multiple proteins essential for the proliferation and survival of cancer cells [75,76,77,78]. GDN contains a benzoquinone moiety which is responsible for (a) redox activity producing superoxide and (b) reactivity with thiols. Importantly, both these interactions contribute to GDN cytotoxicity, however, primarily at high micromolar concentrations [79,80]. For example, Clark and coworkers studied the contribution of ROSs generated by the GDN benzoquinone group to cell death induction in rat pheochromocytoma PC-12 cells [80]. In these experiments, GDN induced ROS production, GSH depletion, and cell death. Addition of NAC at millimolar concentrations reversed all adverse effects including cell death. Protective effects of NAC were attributed to the restoration of GSH synthesis [80]. It is necessary to note that GDN was used at 20 μM concentration [80]. In our laboratory, we performed similar experiments with similar results, but with one significant difference. The addition of NAC led to the quantitative conversion of GDN to the non-cytotoxic GDN–NAC adduct. Therefore, our interpretation of the protective effects of NAC was completely different [55]. 

##### NAC and Cytotoxicity of FCCP

FCCP acts as an H^+^ ionophore and uncoupler of oxidative phosphorylation in mitochondria. It exhibits uncoupling properties starting from 10 nM concentrations in cell-free systems [81]. To achieve an uncoupling effect in intact cells, low micromolar concentrations of FCCP are needed [70]. At high micromolar concentrations, FCCP induces cessation of cell proliferation and cell death, which is accompanied by ROS production and GSH depletion [70,82,83,84]. Further, it was found that NAC at millimolar concentrations in contrast to other ROS scavengers prevented all adverse effects including cell death [82,83,84]. Researchers from the Parks group concluded that NAC’s protective effects are attributed to GSH repletion [82,83,84]. Although we achieved similar results, our interpretation was completely different. We found that NAC formed a noncytotoxic adduct with FCCP, which led to the rapid depletion of free FCCP and thus to the disappearance of its cytotoxic effects [70]. Our results were inspired by the forgotten work of Drobnica and Sturdik, who were the first to describe the formation of adducts between phenylhydrazone derivatives and thiols [85]. 

##### NAC and Cytotoxicity of EF-24

EF-24 is a synthetic fluorinated analogue of curcumin with improved antiproliferative properties against cancer cells both in vitro and in animal models in vivo [86,87,88,89,90]. Anticancer effects of EF-24 are attributed to the suppression of NF-κB activity and by deregulation of oncogenic signalling pathways that include PTEN, Akt, and HIF-1α in cancer cell lines in vitro [87,90,91,92]. However, some authors have suggested that EF-24 induced apoptosis is at least in part redox-dependent [93,94]. Others have reported the anti-cancer activity of EF-24 may be mediated by increased production of ROS [95,96,97]. Importantly, protective effects of NAC, which prevent EF-24-induced cell death, have been attributed to its ability to scavenge ROSs [95,96,97]. The results of our laboratory showed that the addition of NAC to the growth medium containing EF-24 leads to its rapid conversion to a non-toxic EF-24–NAC adduct [71]. Similarly, EF-24 forms an adduct with GSH [71]. 

#### 3.3.4. NAC and Cytotoxicity of Organic Isothiocyanates

Organic isothiocyanates (ITCs) mostly isolated from cruciferous vegetables were reported to exhibit antiproliferative and proapoptotic effects in cancer cells [98,99]. Importantly, in these experiments, cell death was accompanied by ROS production and GSH depletion [100,101,102]. The finding that NAC prevents ROS production, GSH depletion, and cell death is used to support the interpretation that cell death induced by isothiocyanates is related to oxidative stress. Although the mechanism by which NAC protects cells has not been directly studied, its ability to scavenge ROSs or stimulate GSH synthesis is almost always pointed to [100,101,102]. The possibility that NAC directly reacts with isothiocyanates to form the respective adducts and thereby eliminate their cytotoxic effects has not been generally considered, although studies have been published that clearly demonstrate this [103,104]. The fact that organic isothiocyanates can rapidly react with thiols including NAC to form the corresponding adducts was published more than forty years ago [103]. This work was later followed up by Mi and coworkers, who convincingly demonstrated that the formation of an adduct between isothiocyanate and NAC, which occurs in the growth medium, dramatically reduces the uptake of isothiocyanate into cells, leading to the inhibition of all its downstream cytotoxic effects [104].

#### 3.3.5. NAC and Cytotoxicity of Hemin

Only recently, Georgiou-Siafis and coworkers demonstrated that the formation of an NAC–hemin adduct abrogates the cytotoxicity of hemin in human leukaemia cells [105]. Hemine cytotoxicity exhibited similar properties of the abovementioned experimental systems that include ROS production, GSH depletion, cessation of cell proliferation, and loss of cell viability [105,106]. Importantly, all adverse effects can be suppressed by the addition of NAC at millimolar concentrations [105,106]. On a superficial view, it would allow the results to be interpreted as meaning that the protective effects of NAC are due to ROS scavenging and/or repletion of GSH. However, Georgiou-Siafis et al. studied the protective effects of NAC in detail and found that they were associated with reduced intracellular accumulation of hemin [105]. Further analyses have shown that NAC forms an adduct with hemin and that this reaction represents a true protective mechanism of NAC against hemin cytotoxicity [105]. 

### 3.4. NAC as a Direct Modulator of Signalling Pathways In Vitro

In addition to the above protective mechanisms, NAC also affects redox-sensitive transcription factors such as activator protein 1 (AP-1), nuclear factor kappaB (NF-kappaB), and some others through direct reduction of protein disulphide bonds. Thus, NAC indirectly affects the major signalling pathways of cell proliferation and survival, at least in laboratory experiments [69,107]. I will not discuss this protective mechanism of NAC further here, as it is beyond the scope of this review article. Details on this topic can be found in dedicated review articles [12,47,108]. 

### 3.5. NAC Functions as an Efficient ROS Scavenger by Triggering Intracellular Sulphane Sulphur Production In Vitro 

Only recently have Ezerina et al. provided a new explanation for how NAC might function as a potent ROS scavenger [11]. They showed that NAC is converted in the mitochondria of human lung adenocarcinoma H838 cells to form sulphane sulphur species, which are very potent ROS scavengers [11]. However, to what extent this conversion also occurs in other cell lines and to what degree this recently described protective mechanism actually works in laboratory experiments requires further study.

## 4. Why Direct Interaction between NAC and a Cytotoxic Electrophile Is So Rarely Identified as a Mechanism of Protection

The question is how it is possible that the cytoprotective effects of NAC have been attributed to its nucleophilicity in only a few cases, while studies by organic chemists clearly show that thiols, including NAC, react very rapidly with many cytotoxic electrophiles under physiological conditions [57,58,59]. Why we are content with a routine explanation that is generally accepted but often unsupported by detailed experimental evidence and that may in fact contradict the results of other researchers’ studies? In this review, I try to find an explanation for why this is so. I believe that there are several reasons that may lead to incorrect conclusions about the cytoprotective mechanism of action of NAC:(i)The possible interaction between the nucleophilic NAC and the electrophilic cytotoxic agent under study is generally not considered;(ii)The ability of NAC to act as a universal and efficient ROS scavenger is overrated;(iii)The adverse effects of increased ROS production are overestimated;(iv)The NAC concentration used to achieve a protective effect is not taken into account;(v)The apparent similarity between the protective mechanisms involving nucleophile-electrophile interaction and direct or indirect ROS scavenging activity.

The following paragraphs refer to points (i) to (v). Here I will try to use a simplified example to show how incorrect conclusions can be reached, which can then unfortunately appear in the scientific literature.

A studied electrophilic cytotoxic agent induces cell death that is accompanied by increased production of ROS and/or by GSH depletion. All adverse effects, including cell death, can be prevented by the addition of NAC, usually at high millimolar concentrations (Figure 2a,b). Such an effect of NAC is often used as comprehensive evidence that cell death induced by the cytotoxic agent under study is mediated by increased ROS production and/or GSH depletion and that the protective effect of NAC is mediated by scavenging of ROSs and/or GSH repletion via stimulation of GSH biosynthesis. The possibility that NAC could inhibit the cytotoxic effects of an electrophile by reacting with it as a nucleophile and convert it quantitatively into a non-cytotoxic NAC–electrophilic adduct is not considered (Figure 4). This is despite the fact that a number of questions remain unanswered. 

For example, what ROSs are produced in a given experimental system and whether the NAC can actually scavenge them is often not addressed. In most experiments, 2′,7′-dichlorodihydrofluorescein diacetate (H_2_DCFDA) is commonly used to detect ROS generation. However, H_2_DCFDA is non-selective and therefore provides an overall index of oxidative stress without indicating specific ROS [109]. It is often and incorrectly assumed that the NAC is able to effectively scavenge all ROSs. However, as mentioned above, NAC is unable to scavenge either superoxide or H_2_O_2_ [4,5,32,45,46,47]. In addition, the effect of oxidative stress is not correctly assessed and its importance is often overestimated. Indeed, in these experiments, the increased production of ROS is measurable, but it is usually small or moderate and is not enough to kill the cells by itself.

Other open questions concern GSH. Although there is no doubt that NAC serves as a precursor for GSH biosynthesis and can theoretically protect cells from depletion in this way, a deeper analysis is needed. An important and often overlooked aspect of this issue is to address the mechanism by which GSH is depleted. It is essential to find out whether GSH depletion is due to its oxidation to GSSG or whether the GSSG level is affected only marginally during its depletion. Another warning sign that the protective mechanism of NAC may not be related to the stimulation of GSH synthesis is the fact that very high concentrations of NAC, usually 5–10 mM, are required to achieve protection. Indeed, such high concentrations of NAC are superfluous for GSH synthesis stimulation [54,55]. 

In my opinion, the possibility that NAC could inhibit the cytotoxic effects of an electrophile by reacting with it as a nucleophile and quantitatively converting it to a non-cytotoxic NAC–electrophilic adduct should always be considered (Figure 4), especially when the explanation using the previous mechanisms proves to be somewhat problematic in a detailed analysis. The latter mechanism of the cytoprotective effect of NAC apparently has the same concomitant effects: ROS production disappears and GSH synthesis is restored. However, NAC does not actually scavenge ROSs or restore GSH synthesis. It eliminates the cytotoxic agent itself, and therefore neither ROS production nor GSH depletion occur (Figure 4). In these experimental systems, GSH depletion is not caused by its oxidation to GSSG, but by its non-enzymatic interaction with the cytotoxic electrophile to form the GSH–electrophilic adduct(s) (Figure 3). At this point, it is important to realise that GSH is also a potent nucleophile that can react non-enzymatically with a cytotoxic electrophile. This reaction is the real cause of the GSH depletion (Figure 3). It is important to mention here that GSH depletion due to electrophile conjugation does not lead to a massive increase in the GSSG level. High concentrations of NAC are necessary for the reaction shift to the adduct formation side. In practice, this means that the cytotoxic electrophile is quantitatively converted to the corresponding adduct(s) with NAC, which has lost its cytotoxic properties (Figure 4). High concentrations of NAC also prevent the formation of the GSH–electrophile adduct (Figure 4).

The mechanism of protection that includes nucleophile–electrophile interaction can be directly distinguished from the previous ones only by identifying the resulting electrophile–NAC adduct(s) using a suitable analytical method such as liquid chromatography coupled with mass spectrometry (LC/MS). Similarly, to detect the resulting GSH–electrophilic adduct(s), specialised techniques and equipment will be needed, as a reliable but indirect indicator of this protective mechanism is the finding that the addition of NAC to the growth medium reduces the intracellular concentration of the cytotoxic electrophile (its cellular uptake). 

## 5. NAC as Nucleophile Can Prevent Cytotoxicity of Other Electrophiles In Vitro 

By definition, an electrophile is a molecule that in reaction receives an electron pair provided by a nucleophile and thus forms a chemical bond with the nucleophile. It follows that there are a large number of electrophiles, both inorganic (e.g., BF_3_ or Br_2_) and organic (e.g., alkyl halides or acyl halides). This review focuses on organic electrophiles, which are the common subject of toxicological studies.

Organic electrophiles can then be divided according to their origin into exogenous, which can be some environmental pollutants or they are components of the diet, and endogenous, which arise in our cells via enzymatic and non-enzymatic pathways [110]. Importantly, some endogenous electrophiles have even been named reactive electrophilic species (RES) and have been found to be important cellular signalling molecules. A comprehensive review article on this topic was recently published by Parvez et al. [110]. Due to their high chemical reactivity, virtually all exogenous and endogenous electrophiles are cytotoxic if they exceed a certain concentration. As mentioned in foregoing chapters, NAC as a reactive nucleophilic compound can directly interact with a number of cytotoxic electrophiles studied under physiological conditions to form non-cytotoxic adducts. Thus, it is sufficient to know that the cytotoxic chemical studied has the nature of an electrophile. How do we recognise them if we do not have a good knowledge of organic chemistry and the above definition will not help us much in practice? In this regard, the search for so-called electrophilic structural moiety can help us, as their presence in an organic molecule guarantees its electrophilic nature. An overview of the most important electrophilic structural moieties are given in the following paragraphs and Table 2.

α,β-unsaturated carbonyls probably represent the broadest group of electrophilic compounds that readily interact with the thiolate groups of NAC or GSH [56,111, 112]. Although this moiety can be found in a number of synthetic and natural cytotoxic compounds, only a few of them have been shown to interact directly with NAC and/or GSH ([55,64,69], Table 2). However, there remain a plethora of other cytotoxic α,β-unsaturated carbonyls, such as acrolein, acrylamide, 4-hydroxynonenal, henenalin, hypothemycin, tyrosine kinase inhibitors (e.g., osimertinib), etc., useful to (re)consider in relation to their interactions with NAC ([56,111,112], Table 2). 

Molecules whose structural components are conjugated dienes, i.e., two double bonds separated by a single bond, represent another important group of electrophiles that can react directly with NAC. Similarly, although many natural or synthetic cytotoxic molecules contain this moiety, only a few have been shown to interact directly with NAC to form non-cytotoxic adducts ([64,71,105], Table 2). In addition, however, there are a large number of other cytotoxic molecules containing a diene moiety (e.g., retinol) or both a diene moiety together with α,β-unsaturated carbonyls (e.g., callystatin A, trichostatin A), whose interactions with an excess of NAC should be studied in detail.

Organic isothiocyanates belong to a well-studied group of cytotoxic electrophiles with antitumour effects that directly react with GSH or NAC to form the respective adducts [103,104]. Although isothiocyanate-induced cell death is accompanied by increased ROS production, GSH depletion, and NAC is able to prevent all adverse effects, no one has yet attributed these protective effects to its ability to directly or indirectly scavenge ROSs. I hope it stays that way.

Direct reactions between hydrazone derivatives and thiols under near physiological conditions were demonstrated approximately fifty years ago [85]. Consistent with this apparently forgotten work, we have shown, unlike others, that NAC prevents FCCP cytotoxicity by reacting directly with it to form a non-cytotoxic adduct [70]. For this reason, I recommend that the interaction with NAC be carefully investigated for other cytotoxic compounds with a hydrazone moiety (Table 2).

Cytotoxic molecules whose structural component is epoxide or organic acid anhydride are other types of electrophiles that can directly form adducts with NAC. To the best of my knowledge, I do not yet know of any published example where an excess of NAC would prevent their cytotoxic action by forming the appropriate adducts. Even in the case of cytotoxic molecules possessing epoxy or anhydride moieties, it is appropriate to study whether the observed cytoprotective effect of NAC is related to the formation of non-cytotoxic adducts (Table 2).

The mechanism of NAC protection against cytotoxic electrophiles must be studied in more detail before a definitive conclusion is reached. The possibility that NAC as a nucleophile prevents the cytotoxicity of the studied electrophile by direct interaction to form the corresponding adduct should always be considered. Table 2 lists electrophilic molecules whose cytotoxicity is or could be eliminated by direct interaction with nucleophilic NAC.

## 6. Conclusions

A number of cytotoxic agents that are electrophiles induce cell death that is accompanied by ROS production and/or GSH depletion, and all the adverse effects can be prevented by millimolar concentrations of NAC. However, the conclusion that NAC’s protective effect is due to ROS scavenging and or GSH repletion may not be correct. Before reaching a definitive conclusion on the mechanism of the cytoprotective effect of NAC, it is appropriate to investigate whether there is a direct interaction between NAC and the cytotoxic electrophile to form a non-cytotoxic NAC–electrophilic adduct.

## Figures and Tables

**Figure 1 antioxidants-11-01485-f001:**
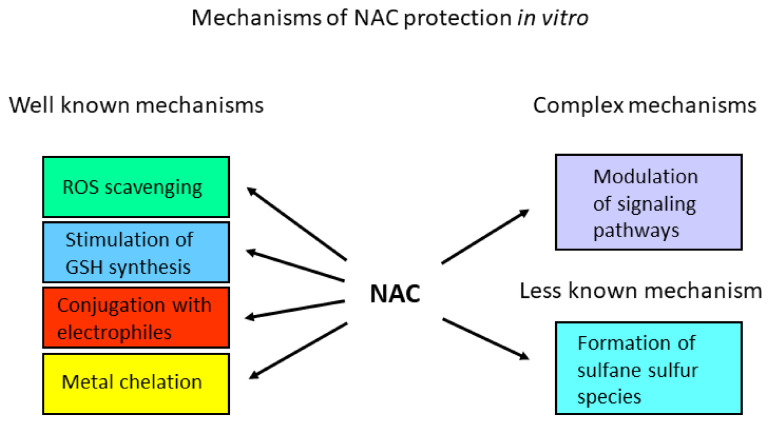
Mechanisms of NAC protection in vitro known so far.

**Figure 2 antioxidants-11-01485-f002:**
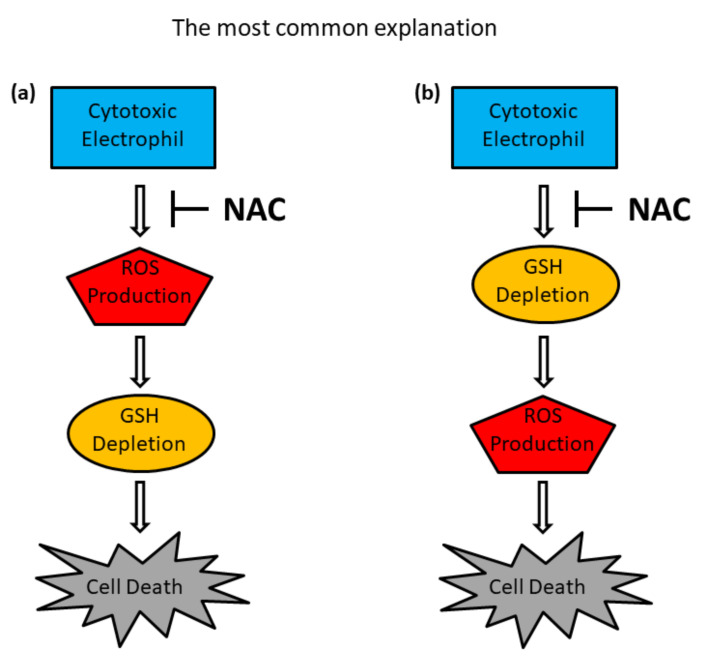
Schematic for putative NAC protection pathways. Cytotoxic effects of an electrophilic agent are accompanied by ROS production, GSH depletion, and eventually result in cell death. Panel (**a**) NAC prevents all adverse effects including cell death by direct scavenging of the ROS. Panel (**b**) NAC prevents all adverse effects including cell death by GSH repletion, i.e., indirect ROS scavenging.

**Figure 3 antioxidants-11-01485-f003:**
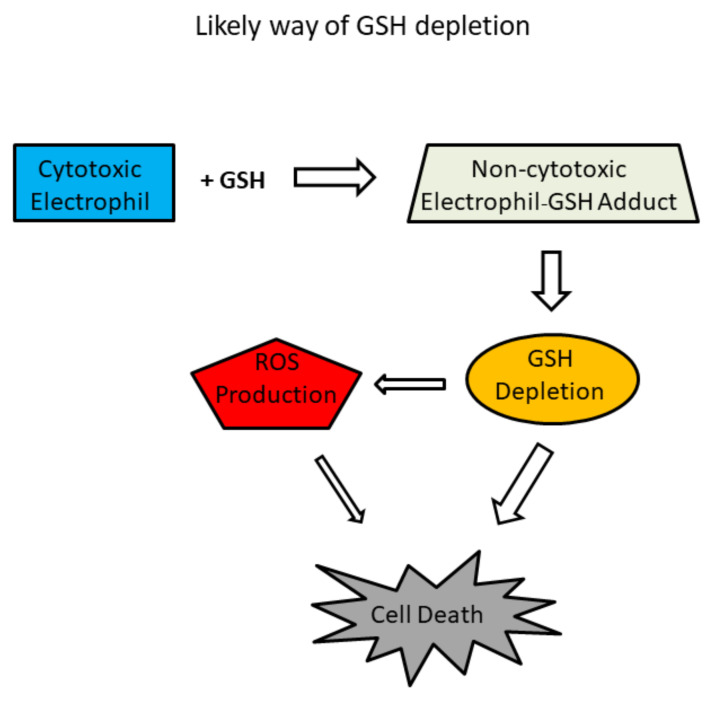
Schematic for predicted GSH depletion. Electrophiles can either directly interact with GSH to form a non-cytotoxic electrophile–GSH adduct or GST (EC 2.5.1.18) can catalyse the conjugation of an electrophile to GSH. This is the true mechanism of GSH depletion, which may be accompanied by increased ROS production and may eventually lead to cell death. Alternatively, the electrophiles can affect a number of signalling pathways to induce increased ROS production.

**Figure 4 antioxidants-11-01485-f004:**
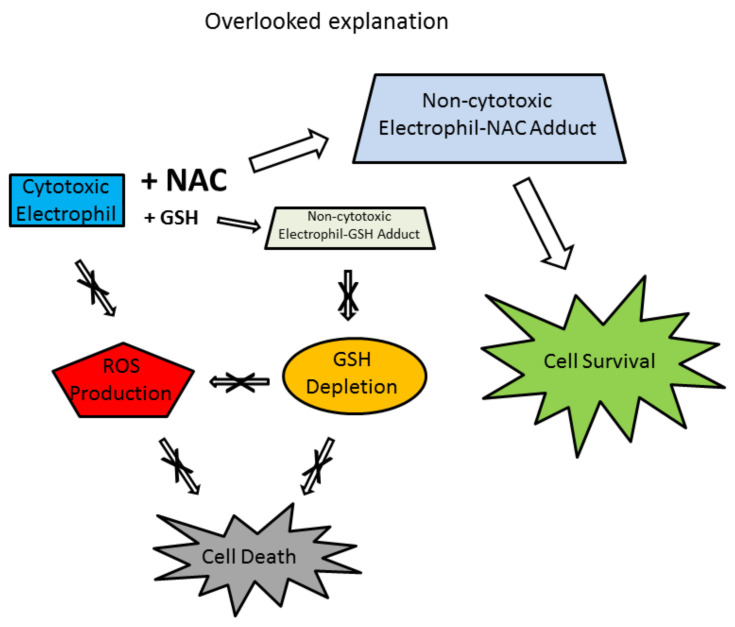
Schematic for the often-overlooked NAC protection mechanism. Electrophiles can directly and quantitatively interact with NAC, which is added in excess, to form a non-cytotoxic electrophile–NAC adduct. This interaction prevents GSH depletion, ROS production, and cell death.

**Table 1 antioxidants-11-01485-t001:** Antioxidants frequently used in laboratory experiments.

Antioxidant	Structure	Protective Effect	Note
Ascorbic acid	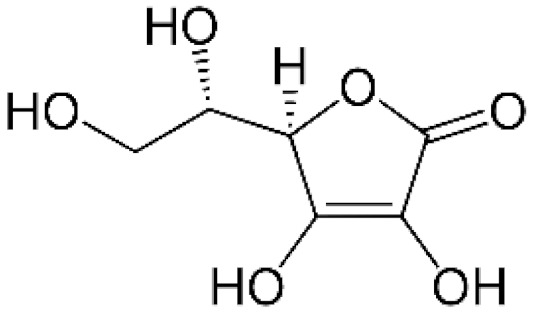	Directly interacts with O_2_•^−^ and H_2_O_2_.	In the presence of iron, it becomes powerful source of ROS.
Ebselen	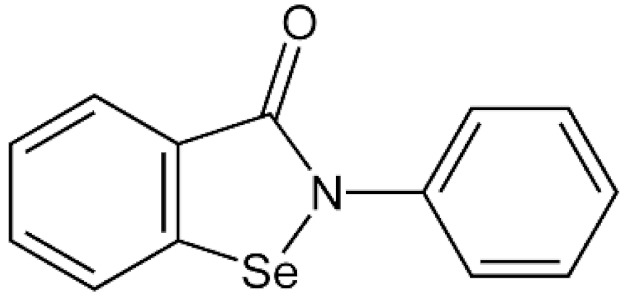	Directly interacts with H_2_O_2_ at low concentrations.	Ebselen becomes powerful source of ROS at high concentrations.
Deferoxamine	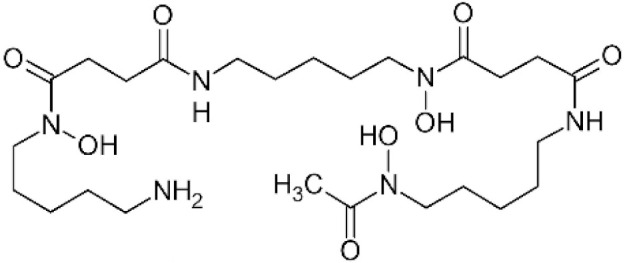	As an iron chelator, inhibits Fenton reaction.	Deferoxamine prevents formation of •OH indirectly.
1,4-Dithiothreitol	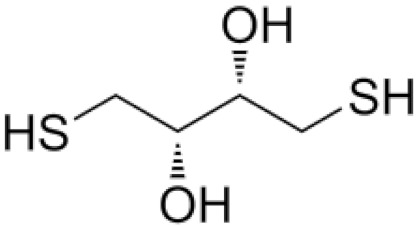	Directly interacts with •OH and reduces disulphide bonds in protein samples.	At neutral and alkaline pH, it autoxidises rapidly. It is used in cell-free extracts.
Glutathione	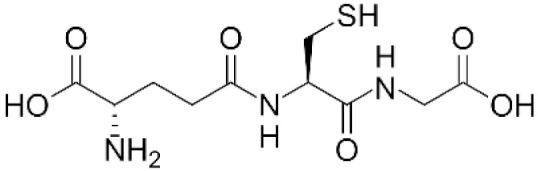	Directly interacts with •OH but not with O_2_•^−^ and H_2_O_2_.	In reactions catalysed by GPx, effectively reduces H_2_O_2_ and ROOH.
Mercaptoethanol	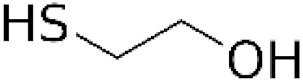	Directly interacts with •OH and reduces disulphide bonds in protein samples.	Due to its relatively high cytotoxicity, it is used in cell-free extracts.
*N*-acetylcysteine	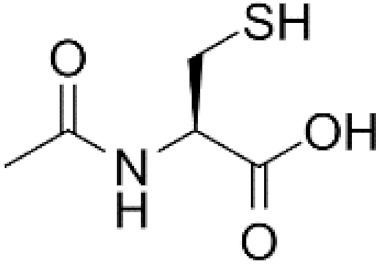	Directly interacts with •OH and HOCl but not with O_2_•^−^ and H_2_O_2_.	Although its application scale is limited, it is overused.
Thiourea	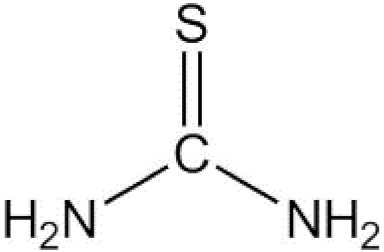	Directly interacts with O_2_•^−^, H_2_O_2_, and •OH.	Although its application scale is very wide, it is used rarely.
Trolox	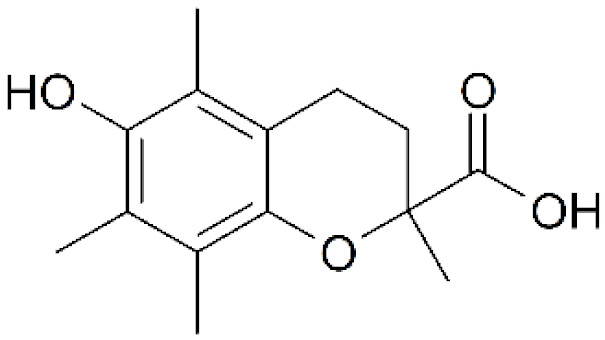	It is used to “repair” a variety of biomolecules damaged by oxidative stress.	A water-soluble analogue of vitamin E.

**Table 2 antioxidants-11-01485-t002:** Electrophiles which cytotoxicity are or could be prevented by nucleophilic interaction with NAC.

Electrophilic Moiety	Structure	Proved Examples	Possible Examples
α,β-unsaturated carbonyls	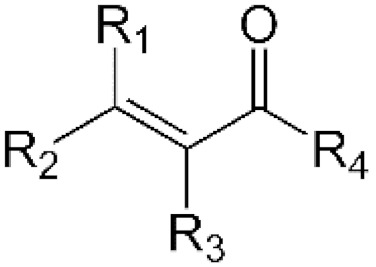	GDN [55], patulin [64], 18β-glycyrrhetinic acid [69]	Acrolein, Acrylamide, 4-hydroxy-2-nonenal, Henenalin, afatinib
Conjugated dienes	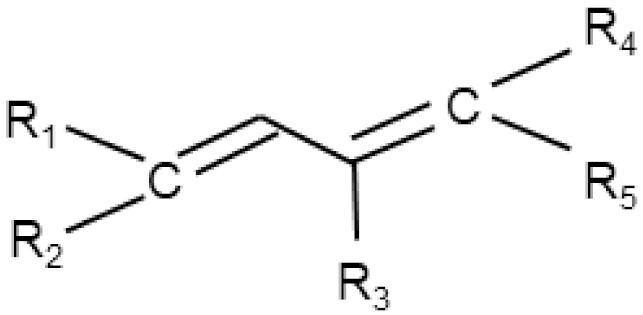	Patulin [64], EF-24 [71], hemin [105]	Retinol, callystatin A, trichostatin A
Organic isothiocyanates	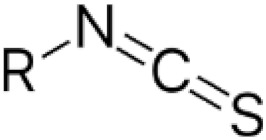	Phenylethyl isothiocyanate [103,104]	Sulphoraphan, benzyl isothiocyanate
Hydrazones	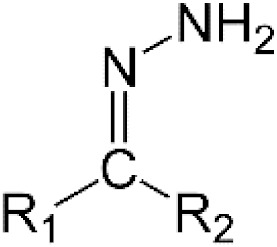	FCCP [70]	CCCP, piperonal ciprofloxacin hydrazone
Epoxides	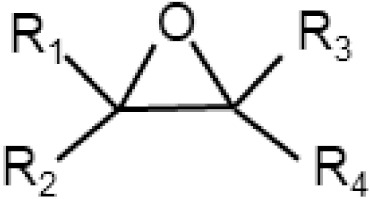	-	Epichlorohydrin, Cantharidin
Organic acid anhydrides	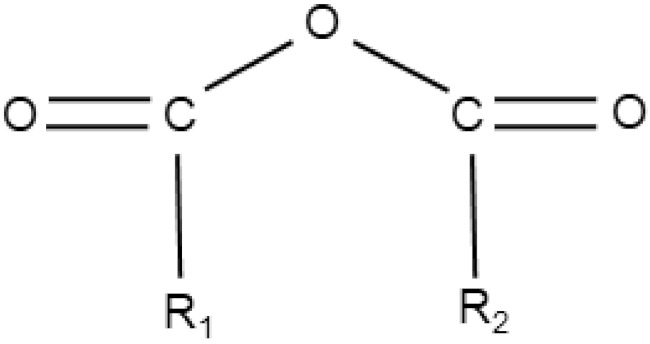	-	Cantharidin

## Data Availability

The data is contained within the manuscript.

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
