# Peer review of "Direct Interaction between N-Acetylcysteine and Cytotoxic Electrophile—An Overlooked In Vitro Mechanism of Protection"

_antioxidants, 2022, doi:10.3390/antiox11081485_

Round 1

Reviewer 1 Report

The manuscript (Antioxidants-1765026) titled "Direct interaction between N-acetylcysteine and cytotoxic electrophile - an overlooked in vitro mechanism of protection" by Petr Mlejnek is an interesting review of the relatively poorly discussed N-acetylcysteine (NAC) protective mechanism. This mechanism is based on the interaction of NAC as a nucleophile compound with cytotoxic electrophiles through the formation of non-cytotoxic NAC electrophile adducts.

In this review, the author makes a critical assessment of the commonly accepted interpretation of the mode of action of  N-acetylcysteine (NAC) as a ROS scavenger and as a cysteine donor to replenish the GSH deficit. In the following sections, the author first presents NAC as a substance with multiple possible applications in clinical practice. To demonstrate the involvement of NAC in modulating ROS levels, he presents the definition of ROS, their source and role in biological processes. Then, the author briefly characterizes different cellular antioxidant systems, and presents the antioxidant properties of NAC, as a direct and indirect scavenger of ROS, as well as a cytotoxicity blocker of electrophiles compounds.

Finally, the NAC protective properties based on its nucleophilicity are discussed in more details by way of a few examples.

In general, the study is well designed, and the structure of the individual chapters is clear and comprehensive. This very valuable review work requires however, some small explanations.

* The authors list several electrophile compounds that may interact with NAC. Some of them are anti-cancer drugs. This raises the question of whether there are any experimental data showing that the addition of NAC to anti-cancer therapy may reduce the adverse toxic effects of these drugs, or whether it will result in a reduction in the therapeutic efficacy of these drugs.

** In the light of these data, can NAC as a substance blocking the cytotoxic effect of some electrophilic substances have a much wider application than so far in toxicology?

This valuable review provide extensive knowledge of the in vitro operation of NAC, therefore I support the publication of this work.

Author Response

Comments and Suggestions for Authors

The manuscript (Antioxidants-1765026) titled "Direct interaction between N-acetylcysteine and cytotoxic electrophile - an overlooked in vitro mechanism of protection" by Petr Mlejnek is an interesting review of the relatively poorly discussed N-acetylcysteine (NAC) protective mechanism. This mechanism is based on the interaction of NAC as a nucleophile compound with cytotoxic electrophiles through the formation of non-cytotoxic NAC electrophile adducts.

In this review, the author makes a critical assessment of the commonly accepted interpretation of the mode of action of N-acetylcysteine (NAC) as a ROS scavenger and as a cysteine donor to replenish the GSH deficit. In the following sections, the author first presents NAC as a substance with multiple possible applications in clinical practice. To demonstrate the involvement of NAC in modulating ROS levels, he presents the definition of ROS, their source and role in biological processes. Then, the author briefly characterizes different cellular antioxidant systems, and presents the antioxidant properties of NAC, as a direct and indirect scavenger of ROS, as well as a cytotoxicity blocker of electrophiles compounds.

Finally, the NAC protective properties based on its nucleophilicity are discussed in more details by way of a few examples.

In general, the study is well designed, and the structure of the individual chapters is clear and comprehensive. This very valuable review work requires however, some small explanations.

Reply:

Thanks to the Reviewer for carefully reading my MS! I am very pleased that you rate it positively.

* The authors list several electrophile compounds that may interact with NAC. Some of them are anti-cancer drugs. This raises the question of whether there are any experimental data showing that the addition of NAC to anti-cancer therapy may reduce the adverse toxic effects of these drugs, or whether it will result in a reduction in the therapeutic efficacy of these drugs.

Reply:

I am not an expert on the clinical application of NAC, however I know it is used to reduce the side effects of cis platinum treatment. 

** In the light of these data, can NAC as a substance blocking the cytotoxic effect of some electrophilic substances have a much wider application than so far in toxicology?

Reply:

Unfortunately, I am not clinically oriented, however, I believe that you are right.

This valuable review provide extensive knowledge of the in vitro operation of NAC, therefore I support the publication of this work.

Reviewer 2 Report

In the review entitled " Direct interaction between N-acetylcysteine and cytotoxic electrophile – an overlooked in vitro mechanism of protection" the author critically examines the existing literature on the antioxidant role of NAC and, from his own data and from the data available in the literature, proposes a mechanism of action of the NAC. He suggests that the cytoprotective effects of NAC in vitro are due to a direct interaction between NAC and the cytotoxic electrophiles.

The whole organization of the review is weak.

Paragraph 2 is a surface synthesis of reactive oxygen species, ROS sources and their biological effects.

Furthermore, it is not clear why the author focuses his attention on antioxidants used in vitro.

In the opinion of this reviewer, the author should frame his review from the very beginning on the use of antioxidants in in vitro research and the problems associated with that use.

In paragraph 3.3 the authors propose the mechanism of action of NAC without providing any supporting research data. This reviewer understands that some information is given in the following paragraphs, but this is a very unique way of proposing a mechanism: first the conclusion then the supporting data!

3.3.1 No information is furnished that patulin interacts with NAC,

Lines 379-381:  the statements reported in these lines are generic and it is not clear what the author refers to.  What does it mean that oxidative stress is not properly assessed?

Lines 398-401 To which experimental systems the author refers.

Author Response

Comments and Suggestions for Authors

In the review entitled " Direct interaction between N-acetylcysteine and cytotoxic electrophile – an overlooked in vitro mechanism of protection" the author critically examines the existing literature on the antioxidant role of NAC and, from his own data and from the data available in the literature, proposes a mechanism of action of the NAC. He suggests that the cytoprotective effects of NAC in vitro are due to a direct interaction between NAC and the cytotoxic electrophiles.

The whole organization of the review is weak.

Reply:

Thanks to the Reviewer for critical comments on my MS! I am very sorry for your negative evaluation of my article. However, while reading your review, I found out that the reason for your negative evaluation of my review article is its misunderstanding. Therefore, I have substantially rewritten the MS to make it easier to understand. In addition, I will try to explain some specific misunderstandings to you. Changes are in cyan.

First, the reason why I wrote this review was as follows. The fact that NAC is generally able to eliminate the cytotoxic effects of electrophilic substances by 3 different mechanisms has long been known. These are: direct scavenging of ROS, indirect effect through stimulation of GSH, and by direct non-enzymatic interaction to form non-cytotoxic adduct NAC-electrophile. Unfortunately, the first two mechanisms are used in the literature to explain the protective effects of NAC without further analysis. And the third option is unjustifiably overlooked. I want that to change! I want to make others think about the real mechanisms of NAC protection and not settle for a generally accepted but often wrong conclusion!

Paragraph 2 is a surface synthesis of reactive oxygen species, ROS sources and their biological effects.

Reply:

Yes, you're right, but that's really not the most important thing in my review. That's why it is.

Furthermore, it is not clear why the author focuses his attention on antioxidants used in vitro.

Reply:

The above-mentioned problem (insufficient analysis of the possible protective mechanisms of NAC) manifests itself especially in laboratory experiments. The Introduction of the article has been rewritten to better explain why I address in vitro issues. Please, see revised MS.

In the opinion of this reviewer, the author should frame his review from the very beginning on the use of antioxidants in in vitro research and the problems associated with that use.

Reply:

However, for the above reasons, I want to focus mainly on NAC. The properties of other antioxidants are only for more complete information of the readers.

In paragraph 3.3 the authors propose the mechanism of action of NAC without providing any supporting research data. This reviewer understands that some information is given in the following paragraphs, but this is a very unique way of proposing a mechanism: first the conclusion then the supporting data!

Reply:

Unfortunately, this is misunderstanding. The fact that NAC can eliminate cytotoxic effect of electrophiles by direct non-enzymatic interaction to form adducts has been known for a long time. I emphasized this fact in the revised version. And I want to point out here that this mechanism should be taken into accountPlease, see revised MS.

3.3.1 No information is furnished that patulin interacts with NAC,

Reply:

At this point I cannot agree with the Reviewer. The fact that patulin interacts with NAC was provided on the page 8, lines 248-250 and reference [61], please look back to the original MS. However, the original sentences have been reworded and extended to better emphasize this fact. Please, see revised MS.

Lines 379-381:  the statements reported in these lines are generic and it is not clear what the author refers to.  What does it mean that oxidative stress is not properly assessed?

Reply:

The relevant part has been rewritten to make it easier to understand. Please, see revised MS.

Lines 398-401 To which experimental systems the author refers.

Reply:

The relevant part has been rewritten to make it easier to understand. Please, see revised MS.

Reviewer 3 Report

Dear Editor,

Please, find below a revision of the manuscript entitled: “Direct interaction between N-acetylcysteine and cytotoxic electrophile – an overlooked in vitro mechanism of protection” submitted to the journal - Antioxidants.

N-acetylcysteine is a derivative of cysteine and modulates the biosynthesis of glutathione. On the other hand, NAC contains the reactive thiol group, thus might participate in antioxidative reactions and play an anti-inflammatory role.

The article is very interesting, and the author has shown an important problem concerning the protective role of NAC against toxic ROS and/or electrophiles.

Therefore, I propose to accept the manuscript with a minor correction.

Comments

In the section 2 of the review article, the author should show the mechanism of the formation of reactive oxygen species. He should describe only those ROS that could be scavenged or generated by N-acetylcysteine.

In the 2.3. section, the author should show rater toxicity mechanisms of the ROS against biological systems instead of well-known as such toxicity.

The issue concerning electrophiles is described poorly. I suggest adding to this article the section on the generation of electrophiles and their harmful effects.

On the other hand, the title suggests the author will describe the mechanism of protection affected by NAC. Thus, maybe in this review article, the author should show the reaction of the NAC with electrophiles and further their scavenging.

In my opinion, the figures presented in the article show only schemas but not mechanisms.

Author Response

Comments and Suggestions for Authors

Dear Editor,

Please, find below a revision of the manuscript entitled: “Direct interaction between N-acetylcysteine and cytotoxic electrophile – an overlooked in vitro mechanism of protection” submitted to the journal - Antioxidants.

N-acetylcysteine is a derivative of cysteine and modulates the biosynthesis of glutathione. On the other hand, NAC contains the reactive thiol group, thus might participate in antioxidative reactions and play an anti-inflammatory role.

The article is very interesting, and the author has shown an important problem concerning the protective role of NAC against toxic ROS and/or electrophiles.

Therefore, I propose to accept the manuscript with a minor correction.

Reply:

I thank the Reviewer for carefully reading the MS. I am very delighted that you rate it positively! I have adjusted the MS according to your suggestions. Changes are in yellow.

Comments

In the section 2 of the review article, the author should show the mechanism of the formation of reactive oxygen species. He should describe only those ROS that could be scavenged or generated by N-acetylcysteine.

Reply:

Thank you for the interesting suggestion. However, this topic has already been covered by other authors. I mainly want to draw attention to the incorrectly assessed protective mechanisms of NAC.

In the 2.3. section, the author should show rater toxicity mechanisms of the ROS against biological systems instead of well-known as such toxicity.

Reply:

Thank you for your opinion, but I see it differently. This is not the point of mine review.

The issue concerning electrophiles is described poorly. I suggest adding to this article the section on the generation of electrophiles and their harmful effects.

Reply:

According to your suggestion, information about electrophiles has been expanded. A citation linking to a comprehensive review on the subject has also been added.

On the other hand, the title suggests the author will describe the mechanism of protection affected by NAC. Thus, maybe in this review article, the author should show the reaction of the NAC with electrophiles and further their scavenging.

Reply:

I understand your comment, but I don't want to go into the detailed mechanism, which by the way is different for each compound. This review article is written to point out that this mechanism is unfairly overlooked as such regardless of the detailed differences.

In my opinion, the figures presented in the article show only schemas but not mechanisms.

Reply:

According to your suggestion, I tried to avoid this naming. Please, see revised MS.

Round 2

Reviewer 2 Report

The revised version of the review article seems much clearer to me.  I only recommend a minor revision of the English because I have noticed minor typos